# Advances in *S. cerevisiae* Engineering for Xylose Fermentation and Biofuel Production: Balancing Growth, Metabolism, and Defense

**DOI:** 10.3390/jof9080786

**Published:** 2023-07-26

**Authors:** Ellen R. Wagner, Audrey P. Gasch

**Affiliations:** 1Laboratory of Genetics, University of Wisconsin-Madison, Madison, WI 53706, USA; 2Great Lakes Bioenergy Research Center, University of Wisconsin-Madison, Madison, WI 53706, USA; 3Center for Genomic Science Innovation, University of Wisconsin-Madison, Madison, WI 53706, USA

**Keywords:** xylose fermentation, signal transduction, yeast, environmental stress response, protein kinase A

## Abstract

Genetically engineering microorganisms to produce chemicals has changed the industrialized world. The budding yeast *Saccharomyces cerevisiae* is frequently used in industry due to its genetic tractability and unique metabolic capabilities. *S. cerevisiae* has been engineered to produce novel compounds from diverse sugars found in lignocellulosic biomass, including pentose sugars, like xylose, not recognized by the organism. Engineering high flux toward novel compounds has proved to be more challenging than anticipated since simply introducing pathway components is often not enough. Several studies show that the rewiring of upstream signaling is required to direct products toward pathways of interest, but doing so can diminish stress tolerance, which is important in industrial conditions. As an example of these challenges, we reviewed *S. cerevisiae* engineering efforts, enabling anaerobic xylose fermentation as a model system and showcasing the regulatory interplay’s controlling growth, metabolism, and stress defense. Enabling xylose fermentation in *S. cerevisiae* requires the introduction of several key metabolic enzymes but also regulatory rewiring of three signaling pathways at the intersection of the growth and stress defense responses: the RAS/PKA, Snf1, and high osmolarity glycerol (HOG) pathways. The current studies reviewed here suggest the modulation of global signaling pathways should be adopted into biorefinery microbial engineering pipelines to increase efficient product yields.

## 1. Microbes Serve as Outstanding Chassis for Biochemical Production

Since the early days of genetic manipulation, microbial engineering through genetic change has revolutionized how chemicals and products of interest to human society are produced. The feasibility of genetic modification, coupled with tractability, ease of culturing, and fast replication rates of the budding yeast *Saccharomyces cerevisiae* and certain bacteria, like *Escherichia coli*, *Lactobacillus*, and others, allow for microbial engineering to produce large yields of designated products in a short period of time. It is now feasible to introduce whole exogenous pathways into *S. cerevisiae* or *E. coli* to produce novel compounds, such as natural plant products like the drugs noscapine and resveratrol [1]. Expressing whole biosynthetic pathways in yeast or bacteria dramatically increases the yield while decreasing necessary resources, allowing for more cost-effective production of bio-products used as flavors, fragrances, and medicines [1]. Another rationale for microbial engineering is to utilize and manipulate an organism’s fundamental physiology. A unique characteristic of *S. cerevisiae*’s biology is its preference for fermentation for energy production, even under aerobic conditions, a characteristic that has been exploited by humans to brew beer, ferment wine, and produce bread for thousands of years [2,3,4,5]. While fermentation is far less energy efficient than cellular respiration, it gives the yeast a competitive advantage in nature: fermentative flux is much faster than respiration, allowing yeast to proliferate faster than other microorganisms. The production of ethanol during fermentation also inhibits the growth of other microorganisms, reinforcing the selective pressures on the yeast to preferentially ferment preferred carbon sources [6]. Once scientists identified yeast as the organism responsible for beer and bread, it was only a matter of time before research techniques advanced enough to manipulate the physiological traits of yeast for industrial purposes.

Modulating innate metabolic pathways like fermentation in *S. cerevisiae* is advantageous because the cell has evolved for millennia to perform that fundamental function. However, this can also become a disadvantage since free-living microbes have evolved to maximize growth when nutrients are plentiful or limit unneeded metabolism and growth to mount a robust stress response under suboptimal conditions (Figure 1). In the remainder of this review, we use “stress response” to refer to any response to an environmental stimulus, whereas “defense” is the aspect of that response that is intended to protect against the stimulus and maintain fitness. An important component of microbial engineering strategies is to drive the pathway flux toward product formation and away from biomass production (i.e., growth) and costly stress defense systems to optimize the product yield (Figure 1) [7,8,9,10]. This requires an understanding of not only the metabolic pathway being engineered but also how that pathway is regulated and integrated with a cellular system.

At the same time, stress tolerance is important for industrial conditions. For example, plant material used to make sustainable fuels must be chemically and/or physically broken down to release the sugars from the biomass, forming a slurry called hydrolysate. The chemicals used to break down lignocellulosic biomass, as well as toxins released from the plants themselves, can inhibit the microorganisms in later steps [11,12,13,14,15,16,17,18,19,20,21,22,23]. Additionally, the composition and concentration of toxins released from the plants change between crop years, depending on the plant’s environmental growth conditions [11,24,25,26,27,28,29]. Ultimately, successful engineering strategies will require the production of stress-tolerant strains in a way that does not compete with cellular resources that are being directed to product formation. A deeper understanding of how cells have evolved to balance growth, metabolism, and stress defense—and how to modulate that through engineering—is required.

## 2. Rapid Growth and Maximal Stress Tolerance Are Competing Interests in Cells

Our understanding of how growth, metabolism, and defense are integrated into cellular regulatory networks is only beginning to emerge, but recent studies have uncovered new insights into the balance between the growth and defense controls. Rapid growth and maximal stress tolerance are competing interests in the cell since both require significant resources to enact. When times are good and nutrients are plentiful, *S. cerevisiae* maximizes its growth rate, but to do so, cells decrease the defense systems to direct resources to biomass production and division. Thus, the fastest-growing cells are typically the most sensitive to acute stress [30,31,32,33,34,35,36]. In response to sudden stress, cells typically decrease their growth rate and transiently arrest their cell cycle while they redirect cellular resources to mounting the stress response, which includes mechanisms to defend against the imposing stress as well as what are likely protective mechanisms against future stresses.

A major component of the *S. cerevisiae* stress response is reorganizing the transcriptome. In addition to specialized responses triggered by specific stresses, stressed yeast mounts a common response to stress. The environmental stress response (ESR) comprises ~900 genes whose expression is altered in response to a variety of stresses, leading to massive physiological changes [37,38,39,40]. The ESR includes ~300 genes whose transcript abundance increases during stress and ~600 genes whose transcript abundance decreases. Induced genes are broadly involved in stress defense processes, including oxidation-reduction balancing, protein folding, the production of defense molecules like trehalose and glycerol, and specific regulators. The transcriptional induction of these genes is controlled by a variety of stress-specific factors in conjunction with the general stress transcription factors Msn2 and Msn4 [37,38,41,42,43]. In contrast, genes repressed in the ESR include genes that normally promote growth, including ribosomal protein (RP) and ribosomal biogenesis (RiBi) genes involved in ribosome production, RNA metabolism, protein synthesis, and cell growth [37].

Activation of the ESR can co-occur with the decreased growth rate of a culture, leading several studies to suggest that the ESR is intimately regulated with, and predictive of, the cellular growth rate [30,44,45,46]. However, work from our lab shows that the ESR is separable from growth and division: the ESR is still activated upon heat or salt stress, even in cells that are already arrested in their cell cycle with low biomass production [34]. Instead, we argue that the dramatic transcriptome changes associated with the ESR serve to accelerate a stress response. The transient repression of ribosome-related and growth-promoting genes during stress helps to redirect the transcriptional and translational capacity toward stress-induced transcripts [34,47,48]. Somewhat counterintuitively, cells that lack repressors of the repressed ESR genes, Dot6 and Tod6, grow well in the absence of stress but acclimate much slower to salt stress [48]. At least part of this mutant effect can be explained by the delayed production of defense proteins: ribosome- and growth-related transcripts stay associated with translating ribosomes in the mutant cells at the expense of stress-induced transcripts, leading to a delay in the production of stress defense proteins [34,48].

The ESR is regulated by multiple upstream signaling pathways, many of which are only activated by specific conditions [33,34,37,38]. Among the best studied of these are the protein kinase A (PKA), Snf1, and high osmolarity glycerol (HOG) pathways, all of which turn out to be important for engineered xylose fermentation (reviewed in more detail below). PKA inhibits the ESR in part by phosphorylating and inhibiting Msn2/4, Dot6/Tod6, and other regulators [49,50]; it also functions to modulate gene expression of specific genes by binding promoters and coding regions via interactions with chromatin proteins or the RNA polymerase [51,52,53,54]. Snf1 and HOG can also modulate downstream ESR regulators, including Msn2/4 and others, both directly and indirectly [41,55,56,57]. Interestingly, several independent studies found that modulating the activity of these broadly acting signaling pathways is necessary to promote robust anaerobic fermentation of xylose in engineered biofuel yeast strains (see below). The remainder of this review will discuss the potential roles of the PKA, Snf1, and HOG pathways in engineering xylose fermentation.

## 3. Engineering Yeast for Ethanol Production from Lignocellulosic Biomass

With its innate proclivity to ferment, *S. cerevisiae* was adopted early for the production of ethanol as a biofuel [58]. While biofuels serve as a renewable fuel source compared to fossil fuels, significant work remains to make their production efficient and sustainable. For example, a major shift in biofuel research was the switch in focus from crop plants, like corn, to lignocellulosic feedstocks as biomass sources for biofuel production [59,60,61,62]. Lignocellulosic feedstocks do not compete with food supply, and they are widely abundant and grow on land less suitable for food crop farming [59,60,61,62]. Another shift in bioenergy research has been to produce higher-energy biofuels, like isobutanol, which is less hygroscopic and has a higher energy density than ethanol, making it more efficient to use in engines [59,62,63,64,65,66,67,68,69]. Engineering robust isobutanol production in yeast is an active area of research that comes with some challenges, explored recently in these extensive studies and reviews: [64,65,67,68,69,70,71,72,73,74,75,76,77].

Maximizing product yields per cell and biomass input requires the conversion of all carbon in the lignocellulosic biomass. This presents a significant bottleneck for efficient biofuel production: several sugars, like the pentose sugar xylose, are prevalent in lignocellulosic biomass, in addition to glucose [78,79]. However, many biofuel organisms, including *S. cerevisiae*, do not natively recognize xylose as a fermentable metabolite, thus limiting product yields when cells leave a large fraction of the sugars unconverted [79]. Thus, a major focus in microbial biofuel research has been the rational engineering of yeast strains to promote robust xylose fermentation.

One strategy has been to learn from other fungi that consume xylose (but often lack other useful traits found in *S. cerevisiae*), and several studies identified genes that, when overexpressed in *S. cerevisiae*, enable or improve xylose metabolism (e.g., *GYC1*, *YPR1*) [80,81,82]. Other studies have used rational engineering and directed laboratory evolution to obtain xylose-fermenting *S. cerevisiae* strains [79,83,84].

Rational design to date starts with the cloning of either xylose isomerase (XI) or xylose reductase, paired with xylitol dehydrogenase (XR/XDH) to convert xylose into D-xylulose for metabolism in the pentose phosphate pathway and central carbon metabolism [79,85,86,87]. However, these enzymes are not enough to support xylose fermentation, with several groups showing that other genetic modifications are required to generate a robust xylose-fermenting strain [79,88,89,90,91,92,93,94,95]. For example, collaborative research in our center, the Great Lakes Bioenergy Research Center, discovered that robust anaerobic xylose fermentation also requires null mutations in the iron–sulfur cluster’s biogenesis chaperone *ISU1*, the stress response MAP kinase *HOG1*, the RAS/PKA inhibitor *IRA2*, and the aldose reductase *GRE3* [96]. Other groups using different strain backgrounds have identified similar suites of mutations [89,90,94]; for example, dos Santos et al. (2016) identified nullifying mutations in *ISU1* and a different member of the Hog1 pathway, *SSK2* [97].

Further work showed that these mutations serve to rewire upstream cellular signaling, pushing the metabolic flow toward metabolism and away from stress responses. Multi-omics studies in our lab discovered major signaling rewiring, causing the simultaneous activation of the growth-promoting signaling pathway, PKA, with the Snf1 pathway, which normally responds to poor carbon sources [98]. PKA is generally active when cells are grown in optimal conditions with glucose as the carbon source, whereas Snf1 is active under suboptimal conditions with non-glucose carbon sources [99,100]. While these pathways share many of the same targets, they have opposing functions and, thus, are not typically active at the same time [99,100]. The physiological impacts of these signaling alterations remain incompletely understood. Furthermore, multiple studies have implicated null mutations in Hog1, which is important both for glucose responses and stress tolerance [101,102,103]. Below, we discuss these pathways and their interplay with growth, metabolism, and defense.

## 4. Key Regulators That Govern Physiological Pathways Rewired for Xylose Fermentation

The PKA, Snf1, and HOG signaling pathways are conserved throughout eukaryotes. All three pathways mediate global changes to the cell in response to a changing environment. While each of these pathways has been extensively studied on their own, how their activity is coordinated with each other and with the cell cycle and metabolism remains poorly understood, thus making it difficult to fully understand their roles in xylose utilization.

### 4.1. Protein Kinase A Pathway

The PKA pathway is one of the best-characterized signaling pathways in eukaryotes. PKA is a growth-promoting kinase, and its main function in microbes is to induce cellular activities that support rapid carbon metabolism and proliferation while simultaneously inhibiting stress responses [104,105,106]. PKA can directly phosphorylate metabolic enzymes to alter biosynthetic flux (e.g., Cdc19, Pyk2, Nth1, and Pfk26), as well as modulate gene expression via the phosphorylation of transcription factors (e.g., Msn2/4, Dot6/Tod6, and many others) and other regulatory kinases (e.g., Yak1) [43,49,104,107,108,109,110]. It is well established that hyperactive PKA prevents growth on non-fermentable sugars via modulating glycolytic enzyme activity, but perhaps also other effects [104]. Strains in which PKA is hyperactive are also sensitive to many types of environmental stressors, such as heat and oxidative stress, consistent with its role in suppressing the stress response [111]. PKA’s role in modulating carbon metabolism is tightly linked with progression through the cell cycle, ensuring that cell functions are supported by the proper nutrients (discussed below). Thus, the PKA pathway must be highly regulated to ensure it is activated appropriately for the conditions.

More recent evidence indicates that PKA may have different effects depending on how it is activated. PKA is a heterotetrameric enzyme composed of two catalytic subunits (encoded by the *TPK1/2/3* genes) and two regulatory subunits (encoded by the *BCY1* gene; Figure 2) [111,112,113]. PKA is thought to be inactive via association with Bcy1 but can become active when cAMP binds Bcy1, releasing the catalytic subunits [99,105]. cAMP is produced by the adenylyl cyclase Cyr1 [114,115,116], which, in yeast, is regulated by two different upstream branches, both activated by the presence of glucose in the environment and the cell. One branch contains the transmembrane G-protein-coupled receptor Gpr1 that binds to external glucose and relays the signal to Cyr1 via Gpa2 (Figure 2) [99]. The second branch involves Ras1/2 GTPase proteins that activate Cyr1 for cAMP production. The guanine nucleotide exchange on Ras1/2 is stimulated by the guanine exchange factors (GEFs) Cdc25 and Sdc25 (Figure 2), which are activated by glucose but do not directly sense glucose [117,118,119]. It is hypothesized that the phosphorylation of glucose to glucose-6-phosphate and the subsequent acidification of the cytosol activates the GEFs, but the exact mechanism remains unknown [120,121,122,123]. This process is inhibited by the GTPase-activating proteins Ira1/2 that convert Ras1/2 to the inactive GDP-bound form [124,125,126,127]. cAMP concentrations are also regulated by a feedback mechanism composed of PKA and the phosphodiesterases (PDEs) Pde1/2, which degrade cAMP into AMP (Figure 2) [128,129,130,131].

There are multiple modes of regulating the RAS/PKA pathway apart from cAMP production. A particularly interesting mode that remains poorly understood is through spatial control. Each of the yeast PKA subunits has its own localization patterns under varying conditions [132,133,134]. However, how this localization is regulated remains incompletely known. Higher eukaryotes have a variety of A-kinase anchoring proteins (AKAPs) that bind the regulatory subunit to control the PKA’s subcellular localization [135]. AKAPs have tissue-specific expression and regulate PKA–substrate interactions [135,136,137,138,139,140]. They have also been described to form signalosomes, which produce micro-environments of PKA, PKA substrates, PDEs, and/or other upstream components of the PKA pathway [141,142]. This is predicted to bring PKA in contact with the substrates and quickly and dynamically regulate PKA activity via cAMP abundance [139,141,142,143,144,145]. While yeast does not possess recognizable AKAP orthologs, several functional analogs have been proposed [146,147,148]. For example, Tpk1 nuclear localization is dependent on the presence of Bcy1 [133], supporting the possibility that Bcy1 either acts as an AKAP or interacts with other AKAP-like proteins. Previous work in our lab found that adding a C-terminal tag to Bcy1 in a strain engineered for xylose metabolism is enough to enable rapid anaerobic xylose fermentation in the absence of growth [98]. This result suggests that Bcy1 has a more nuanced role in PKA regulation than simply binding and inhibiting its activity. Due to its broad roles in regulating carbon metabolism and cell growth, it is no surprise that PKA plays an important role in modulating xylose utilization.

Remarkably, the mode through which PKA is activated impacts xylose fermentation capabilities. Deletion of *IRA2*, in the context of xylose metabolism enzymes and *ISU1* deletion, promotes rapid anaerobic xylose fermentation and growth [96,98]. This is mediated through up-regulated PKA since blocking PKA with specific inhibitors prevents both anaerobic growth and xylose metabolism [98]. Consistent with the requirement of PKA activity, anaerobic xylose fermentation is also enabled by *BCY1* deletion; however, in this case, anaerobic growth on xylose is blocked despite robust fermentation [98,149]. Thus, the Bcy1 regulatory subunit is important for coupling PKA-dependent growth and metabolism. Recent work from our lab shows that this coupling may have to do with PKA-dependent phospholipid metabolism: anaerobic xylose growth and metabolism could be recoupled in a *bcy1∆* strain through directed evolution. The evolved strain carries mutations in the PKA subunit, *TPK1*, and a regulator of phospholipid metabolism, *OPI1*, among other mutations, and has altered phospholipid profiles [149]. While further research will be required to fully elucidate the cellular mechanisms at play, these results show that PKA is intimately coordinated with diverse physiological processes, and engineering approaches will need to consider that coordination for successful strategies.

### 4.2. Snf1 Pathway

Similar to PKA, Snf1 is part of a multi-protein, nutrient-sensing complex that reorganizes the metabolism in the presence of alternative carbon sources, although it also has separable roles in the stress response [150,151]. As glucose is depleted from the environment, Snf1 becomes active to prepare the cell to switch from fermentative to respiratory metabolism, called the diauxic shift [152,153,154,155,156]. During this time, the cell undergoes massive transcriptional alterations. Like PKA, Snf1 interacts with a broad set of protein targets, impacting physiological processes from carbon metabolism and gene expression to intracellular trafficking and cell cycle progression [100]. Perhaps its best-characterized function is modulating gene expression related to carbon source-dependent gene de-repression. Snf1-dependent phosphorylation of transcription factors can be activating (as for the alternative carbon source factors Adr1, Sip4, and Cat80) or inhibitory (e.g., the glucose repression factor, Mig1/2) [99,100]. Additionally, Snf1 can directly regulate chromatin accessibility and the transcriptional machinery. By phosphorylating histone H3, Snf1 recruits the SAGA complex for histone H3 lysine 4 (H3K4) acetylation [99,100]. Other work has shown a direct interaction between Snf1 and the RNA polymerase II holoenzyme, suggesting Snf1 regulates RNA Pol II activity [157,158].

The Snf1 holoenzyme contains three subunits: Snf1 is the α-subunit, which functions as the catalytic kinase; Snf4 functions as the γ-regulatory subunit; and the regulatory ß-subunit that modulates substrate interactions and complex localization is supplied as one of three proteins: Gal83, Sip1, or Sip2 (Figure 3) [159,160,161,162,163,164]. In the presence of glucose, Snf1 is inactive via autoinhibition and the Pma1-regulated intracellular pH [165]. When glucose is depleted from the environment, ADP levels rise and bind Snf4, causing a conformational change that, in turn, protects Snf1 in its active state [163,166]. During this time, Snf1 is also phosphorylated on threonine 210 (Thr210) in its activation loop with Snf4, protecting the residue from dephosphorylation. When the glucose concentration increases, the Snf1 activating protein, Std1, aggregates in puncta [167], while the Glc7-Reg1 protein phosphate complex is activated by hexokinase (Hxk2) to dephosphorylate and inactivate Snf1 [154].

Even though Snf1 is best known for its role in glucose-responsive de-repression, additional functions outside of the central carbon metabolism are being established [151]. Snf1 was shown to phosphorylate and activate the ESR transcription factor Msn2 [37,55,168,169,170] and respond to a variety of stressors including cadmium, hygromycin B, hydroxyurea, selenite, iron, heat, oxidative stress, sodium toxicity, and ER stress [56,170,171,172,173,174,175,176,177,178,179,180]. Snf1 also plays a role in cell cycle regulation (see more below) and cellular aging, where in yeast, it is required to establish chronological aging in cells that have exhausted their replicative age [152,181,182,183].

As the PKA and Snf1 pathways both respond to a carbon source, it is perhaps not surprising that they can regulate each other. PKA controls the localization of the Snf1-Sip1 complex [184] and regulates one of the kinases that phosphorylates and activates Snf1 [185]. In return, Snf1 regulates PKA activity by phosphorylating and inhibiting the adenylyl cyclase [186]. Clearly, significant crosstalk between the Snf1 and PKA pathways exists [151]. The question of how these two opposing pathways are simultaneously activated for xylose fermentation remains unknown.

### 4.3. High Osmolarity Glycerol Pathway

While the PKA and Snf1 pathways respond to nutrients, the high osmolarity glycerol (HOG) pathway is best known for sensing and responding to environmental stressors, particularly changes in osmolarity. The primary effector of the HOG pathway is the mitogen-activated protein kinase (MAPK) Hog1 [103]. After osmotic stress, Hog1 becomes active by one of two upstream branches, which themselves have multiple components [103]. The Sln1 branch is composed of a MAPK signaling cascade, where the transmembrane osmosensor, Sln1, leads to downstream activation of MAP3K Ssk2/22 [187,188,189]. Ssk2/22 phosphorylate and activate the MAP2K Pbs2, which then phosphorylates and activates Hog1 [187,190,191,192]. The Sho1 branch is more complex and regulates two separate physiological responses: osmotic stress adaptation and filamentous growth. Like Sln1, Sho1 is a transmembrane osmosensor that interacts with two other transmembrane osmosensors, Msb2 and Hkr1, through a mechanism that is not fully understood. Through subsequent activation steps (Figure 4), the Sho1 branch converges with the Sln1 branch on phosphorylating and activating Pbs2, which proceeds to activate Hog1 [193]. Once active, a significant portion of Hog1 translocates to the nucleus to alter gene expression and promote the accumulation of intracellular glycerol that balances the osmolarity between the cell and environment [103,194].

The pathway can be inhibited by negative feedback. When the osmotic balance is reached between the cell and the environment, the osmosensors stop relaying a response, leading to the dephosphorylation and inactivation of Pbs2 and Hog1 [190,195,196,197]. Additionally, active Hog1 autoregulates its own pathway by phosphorylating players in the Sho1 branch, disrupting signaling through that branch and thus reducing Hog1 activity [198,199,200].

Similar to the PKA and Snf1 pathways, Hog1 phosphorylates transcription factors to modulate gene expression [201,202,203]. Other methods of transcriptional and post-transcriptional regulation include recruiting transcriptional machinery to chromatin, altering nucleosome positioning, and modifying mRNA stability and transport from the nucleus [103,204,205,206,207]. Hog1 can also directly regulate glycerol biosynthetic enzymes and transporters, which supports the short-term acclimation to osmotic stress [188,196,198,208,209,210,211,212].

Although typically thought of as an osmotic stress regulator, Hog1 can also be activated in response to a variety of environmental changes, several of which connect the Hog1 and Snf1 pathways. Hog1 is activated under both glucose stimulation and starvation in an Snf1-dependent manner that also impacts lipid signaling at the Golgi [101,102]. In contrast, Hog1 activation during ER stress is inhibited by Snf1 activity [171,213]. These studies highlight the complex nature of Hog1 activation and its requirement for competing metabolic processes. In fact, past work in our lab found that improving the stress tolerance in an engineered biofuel yeast strain by maintaining a wildtype *HOG1* reduced the specific xylose consumption rate compared to a *hog1∆* mutant [214]. Thus, engineering an optimal strain for biofuel production and tolerance of industrial conditions will likely require intimate tinkering of multiple signaling pathways.

### 4.4. Cell Cycle Regulation by PKA, Snf1, and HOG Pathways

The cell cycle is closely coordinated with the response to nutrients and stressors. During optimal conditions, the metabolism of nutrients provides the cell with basic resources to support DNA synthesis, rapid changes in gene expression, and mass accumulation. There are several checkpoints throughout the cell cycle to prevent progression if resources are unavailable or in the event of extreme stress in which cells often arrest. With their broad roles in regulating cellular physiology in response to nutrients and stress, it is not surprising that the PKA, Snf1, and HOG pathways contribute to cell cycle regulation.

While PKA and Snf1 have opposing roles in carbon signaling, they both exert positive control over the growth and division. PKA induces the expression of ribosomal protein genes and increases their translation [49,99,215,216]. This is hypothesized to affect the critical size a cell must reach before commitment to the cell cycle [215,217,218,219,220,221,222,223]. Snf1 has been reported to localize to the bud neck during mitosis, promote a proper mitotic spindle arrangement, and regulate the expression of G1-specific genes [224,225,226,227,228]. As we continue to obtain a better understanding of these two pathways, it is very likely that more detailed roles for PKA and Snf1 in cell cycle regulation will be uncovered.

Since stress can have dramatic negative effects on a cycling cell, Hog1 regulates arrest at several stages throughout the cell cycle. Hog1 can cause transient cell cycle arrest if it is activated for a short period of time or can lead the cell to apoptosis if the stress is sustained for an extended time [103,187,194,229,230]. Just as Hog1 modulates physiology via multiple mechanisms in response to stress, it also initiates cell cycle arrest by several methods: First, Hog1 delays the expression of cell phase-specific transcripts to prevent progression. Second, Hog1 directly phosphorylates cell cycle regulatory proteins to modulate their activity, thus preventing progression [231,232,233,234,235,236,237,238,239,240]. Thus, proper Hog1 activity to promote cell cycle arrest in response to stress is crucial for cell survival.

## 5. Future Prospects

Much remains to be understood in terms of how cells normally integrate signaling pathways to control cellular systems and, in turn, how to modulate that integration for desired engineering solutions. The engineering of anaerobic xylose fermentation in *S. cerevisiae* showcases the importance of cellular rewiring but also highlights the challenges ahead for directed engineering. For example, much of our knowledge of the cellular rewiring that enables anaerobic xylose fermentation was discovered by studying mutations that emerged through the laboratory evolution of novel metabolic traits. As the field learns more about how cells have naturally evolved to integrate many signaling systems, the ability to engineer those outcomes for industrial use will also advance.

## 6. Conclusions

Recent work on engineering yeast for xylose fermentation has pointed to the importance of global signaling pathway integration. Research from our lab and others found that modulating the activity of the PKA, Snf1, and Hog1 pathways is required for efficient xylose fermentation, in addition to minimal genetic engineering with xylose metabolic genes. While this review summarized where xylose fermentation in yeast currently stands, it is evident there still remains much to uncover about the interconnectedness of the PKA, Snf1, and Hog1 pathways and how balancing their activity levels impacts growth, metabolism, and stress defense under industrial growth conditions.

## Figures and Tables

**Figure 1 jof-09-00786-f001:**
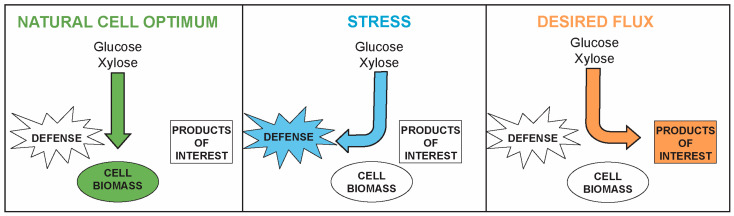
Overview of different cellular regimes that prioritize growth, stress defense response, or engineered metabolic flux in biofuel-producing microorganisms. See text for details.

**Figure 2 jof-09-00786-f002:**
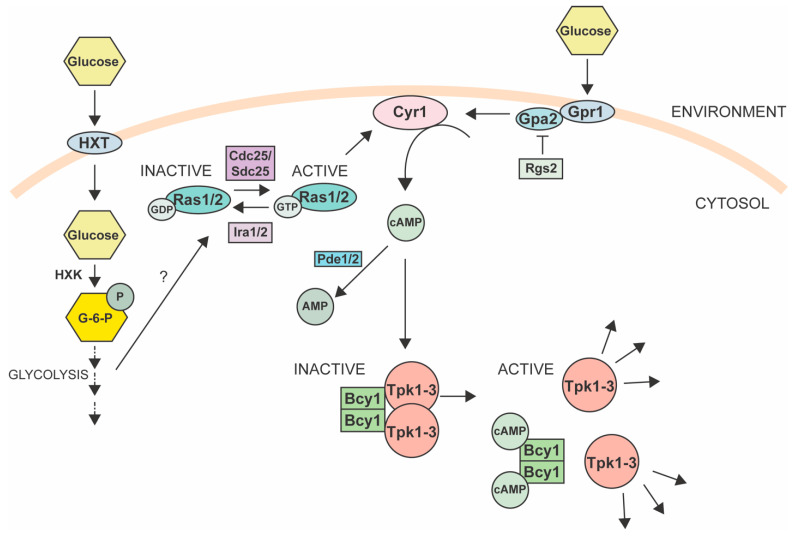
The Protein Kinase A pathway. A simplified view of the RAS/PKA pathway in yeast. HXTs, hexose transporters; HXK, hexokinase. See text for details [99].

**Figure 3 jof-09-00786-f003:**
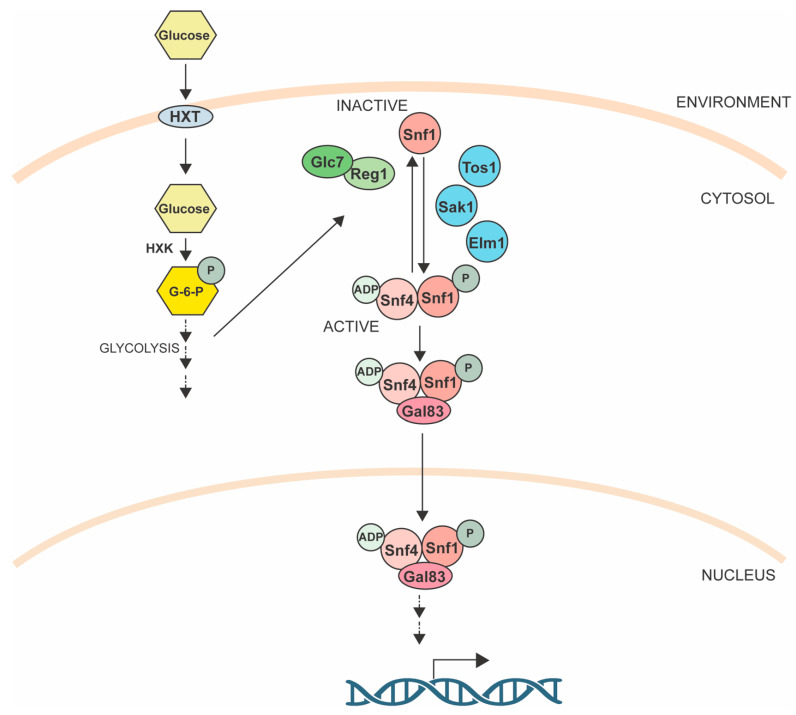
The Snf1 pathway. The Snf1 complex can be regulated by one of three kinases (Tos1, Sak1, or Elm1). Active Snf1 complex translocates into the nucleus to modulate gene expression. When glucose concentration increases, the Glc7-Reg1 protein phosphatase complex is activated by hexokinase (HXK) to dephosphorylate and inactivate Snf1. See text for details [154].

**Figure 4 jof-09-00786-f004:**
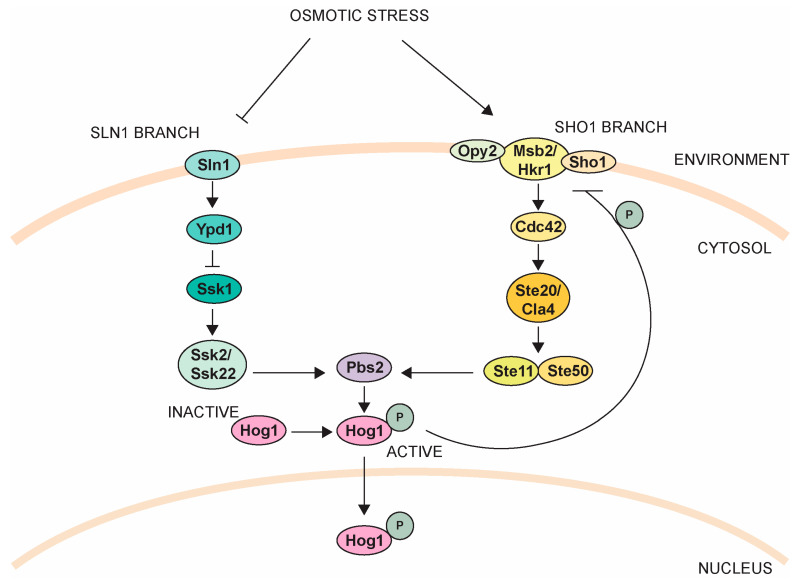
The high osmolarity glycerol pathway. See text for details [103].

## Data Availability

Data sharing not applicable.

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
