# Peer review of "Advances in S. cerevisiae Engineering for Xylose Fermentation and Biofuel Production: Balancing Growth, Metabolism, and Defense"

_jof, 2023, doi:10.3390/jof9080786_

Round 1

Reviewer 1 Report

This review paper focused on a very interesting topic: the correlations among sugar utilization, molecular networks, and stress response in biofuel production by budding yeast. Particularly, this manuscript reviewed in detail the signal pathways related to xylose metabolism and stress defense, which will attract much audience. Only two questions for the authors:

1. Figure 1 is not easy to be understood. What’s the correlation among the three diagrams.

2. How to define “defense”? What’s the difference between “defense” and “stress response”?

Reviewer 2 Report

This article reviewed the adavnces in Saccharomyces cerevisiae for xylose fermentation, including balancing, growth, metabolism, and defense of this fungus. Enabling xylose fermentation in S. cerevisiae requires the introduction of several key metabolic enzymes but also the regulatory rewiring of three signaling pathways at the intersection of growth and stress defence responses. The manuscript is well prepared and written; however, some points need to be considered by the authors.

1.     Please include the terms “Advances” and “xylose fermentation” in the title.

2.     Line 18, Check the word “showcasing”.

3.     What this review can provide to the readers in the field of biorefinery? Please add the reply as the last sentence in the abstract.

4.     The last keyword “environmental stress response Protein Kinase A” need to be shortened.

5.      Change the term “plant biomass” to “Lignocellulosic biomass” in the whole text.

6.     Please, cite the references for Figures 2,3, and 4.

7.     Please include the conclusion section after future prospects. 

Reviewer 3 Report

The review systematically summarizes the strategies that can be employ to biofuel synthesis in S. cerevisiae by balancing existing bottleneck problems. The authors proposed reasonable prospects for biofuel production by balancing growth, metabolism and defense mechanism of cell. The review is well written and organized. I have two minor comment for authors to consider:

a) The title seems misleading. The most of the review is written considering S. cerevisiae as a model system. However, in title it says microbial engineering which is a very broad term. I will suggest to include S. cervisiae instead of microbial engineering.

b) There are two many references. The authors can consider to include only the latest reviews/research articles.

Round 2

Reviewer 2 Report

Th manuscript can be accepted in its current format